# Human emotional odours influence horses' behaviour and physiology

**Plotine Jardat**[1,2*], **Alexandra Destrez**[3], **Fabrice Damon**[3], **Noa Tanguy-Guillo**[2], **Anne-Lyse Lainé**[2], **Céline Parias**[2], **Fabrice Reigner**[4], **Vitor H. B. Ferreira**[2], **Ludovic Calandreau**[2], **Léa Lansade**[2*]

**1** Institut Français du Cheval et de l'Equitation, Pôle développement, Innovation et Recherche, Nouzilly, France, **2** INRAE, CNRS, Université de Tours, PRC, Nouzilly, France, **3** Development of Olfactory Communication and Cognition Laboratory, Centre des Sciences du GoÛt et de l'Alimentation, Institut Agro Dijon, CNRS, Université de Bourgogne-Franche-Comté, Inrae, Dijon, France, **4** UEPAO, INRAE, Nouzilly, France

\* plotine.jardat@ifce.fr (PJ); lea.lansade@inrae.fr (LL)

## Abstract

Olfaction is the most widespread sensory modality animals use to communicate, yet much remains to be discovered about its role. While most studies focused on intraspecific interactions and reproduction, new evidence suggests chemosignals may influence interspecific interactions and emotional communication. This study explores this possibility, investigating the potential role of olfactory signals in human-horse interactions. Cotton pads carrying human odours from fear and joy contexts, or unused pads (control odour) were applied to 43 horses' nostrils during fear tests (suddenness and novelty tests) and human interaction tests (grooming and approach tests). Principal component analysis showed that overall, when exposed to fear-related human odours, horses exhibited significantly heightened fear responses and reduced interaction with humans compared to joy-related and control odours. More precisely, when exposed to fear-related odours, horses touched the human less in the human approach test (effect size: Rate Ratio(RR)=0.60 ± 0.24), gazed more at the novel object (RR = 1.32 ± 0.14), and were more startled (startle intensity – Cohen's d = −0.88 ± 0.39; and maximum heart rate – Cohen's d = 1.16 ± 0.47) by a sudden event. These results highlight the significance of chemosignals in interspecific interactions and provide insights into questions about the impact of domestication on emotional communication. Moreover, these findings have practical implications regarding the significance of handlers' emotional states and its transmission through odours during human-horse interactions.

## Introduction

Olfaction, one of the most primitive senses, is probably the most widespread sensory modality used by animals to communicate. Evidence of communication through

**Data availability statement:** Data and code generated and analysed in this study have been deposited at https://doi.org/10.57745/T2RTDR and are publicly available.

**Funding:** This study was granted by ANR Emodour (grant number ANR-23-CE20-0033) and Institut Français du Cheval et de l'Équitation (IFCE, grant number 32001331 Cognition Emotion). PJ was funded by the French Ministry for Research. These funding sources had no role in the study design, data collection and analysis, decision to publish or preparation and submission of the manuscript.

**Competing interests:** The authors declare no competing interests.

olfaction has been found across most animal groups [1], from insects to mammals, serving a wide range of biological functions. The most essential function is reproduction, where olfaction plays a crucial role, from sexual attraction (e.g., in snakes [2]) and mate recognition (e.g., in amphibians [3]) to mother-offspring recognition (e.g., in degu [4] or in cats [5]), highlighting the importance of chemical communication for fitness and survival. Olfaction is also involved in resource finding, with pheromone trails produced by ants to guide conspecifics toward food resources [6], and in spatial orientation, with wolf signalling biological fields through urinary marking [7]. Olfaction could also be involved in emotional communication, as odorous compounds are secreted in situations of danger, which are often associated with fear. For example, mustelids secrete sulphurous compounds when encountering predators [8]. Similarly, odours produced by decaying sea lampreys are avoided by other sea lampreys [9], which could be explained by fear being triggered in the recipients of this smell leading to avoidance of the related danger. This transmission of emotional information between conspecifics through olfaction has been explored specifically in a few studies. In cows and pigs, increased levels of stress (cortisol response, latency to feed) were observed when animals were exposed to urine from stressed conspecifics [10,11].

In humans, recent research into olfactory communication of emotions has highlighted the role of sweat produced by the apocrine glands in the armpits as carriers of emotional information [12,13]. Compounds present in sweat, such as adrenaline, androstadienone, or hexadecanoic acid have been suggested as potential carriers of this emotional information [14–16]. More than forty studies have demonstrated that such odours can influence the emotional state of the recipient [12]. For instance, chemosignals associated with fear or joy have been found to elicit corresponding fearful or joyful facial expressions in individuals exposed to them [14,17].

Most studies exploring the role of olfaction in social interactions have focused on intraspecific interactions. However, recent studies suggest that olfaction could play a role in interspecific emotional contexts as well. For example, dogs seem to perceive and react to human odours related to stress [18] or fear [19] and they show fewer interactions with a stranger and higher heart rates after smelling human odours related to fear than odours related to joy [20]. Mice and cows also appear to discriminate human odours associated with different emotional states [21], but to date the role of olfaction in interspecific interactions remains insufficiently understood. Further studies on domestic animals other than dogs would allow to better understand the phylogenetic extent of these phenomena and the impact of domestication on cognitive capacities.

Therefore, the aim of the present study was to further explore the role of emotional olfactory signals in interspecific interactions, and more precisely in human-animal interactions. Specifically, we studied horses and their reaction to human emotional odours. Indeed, horses have a presumably developed sense of smell [22] characterized by a convoluted olfactory bulb [23], and olfaction seems important for social interactions among horses [24,25]. Moreover, horses possess advanced cognitive skills related to humans. For example, they can recognize individual human faces in

pictures, remembering them for at least six months [26,27], and they appear to detect our intentions [28]. They also perceive and interpret human emotions, reacting negatively to human voices and faces of anger compared to joy [29,30], and adapting their behaviour toward a human according to emotional cues they previously perceived [31]. They also perform social referencing based on human emotional cues [32], and they show cognitive integration of human emotional signals from different modalities [33–35]. Moreover, recent studies revealed that horses can discriminate human body odours from contexts of fear and joy [36,37] and that they may perceive human joy odours as positive [37].

In this study, we tested whether horses' behaviour and physiology can be influenced when they are exposed to odours emitted by humans feeling fear or joy. We first collected human armpit odours on cotton pads in contexts of joy and fear (volunteers watching horror and joyful videos). A control odour was also included, consisting in unused cotton pads. Then we conducted a series of tests with horses to evaluate whether these odours, when applied to the nostrils, modulate the behaviour and/or the physiology of horses. We used a protocol previously employed to assess the impact of odours (essential oils) on fear responses (novel object test and suddenness test) and on reactions toward humans (grooming test and human approach test [38–41]). Our hypothesis was that horses' behaviour and physiology would vary according to the odour they were exposed to. Specifically, we expected horses to show higher fear responses and to be less prone to interact with humans when smelling the odour from the fear context compared to the control odour and the odour from the joy context, and to show lower fear responses and be more prone to interact with humans when smelling the odour from the joy context, compared to the control odour and the odour from the fear context.

## Methods

### Ethics statement

This study was reported in accordance with ARRIVE guidelines [42]. It was approved by the Val de Loire Ethical Committee (CEEA VdL, Nouzilly, France, authorization number CE19—2022-1511-2). Animal care and experimental treatments complied with the French and European guidelines for the housing and care of animals used for scientific purposes (European Union Directive 2010/63/EU) and were performed under authorization and supervision of official veterinary services (agreement number F371752 delivered to the UEPAO animal facility by the veterinary service of the Département d'Indre et Loire, France). The horses were not food deprived during the experiment and did not undergo any invasive procedures.

Human participation in the experiment was carried out according to the Declaration of Helsinki and was approved by the Institutional Review Board of the University of Tours (authorization number 2022−029). All participants were fully informed about the general aims and methods of the study, and they provided written informed consent for the collection of samples as well as their use in the experiment.

Data and code produced and used in this experiment are available in the following repository: https://doi.org/10.57745/T2RTDR

### Human odour collection

Human odours corresponding to fear and joy contexts were collected by a method identical to that described in [37]. Human axillary sweat odour was collected from 30 adult participants (8 males and 22 females) who volunteered to take part in the experiment. The recruitment period was between 20/11/2022 and 07/03/2023; participants gave written informed consent for participation in the study and use of the collected material for the experiment. They were asked to follow dietary restrictions and use hygiene products provided by the experimenters (identical to [37]). For two days before sweat collection, they had to abstain from consuming certain produces known to affect body odours (chili pepper, spices, blue cheese, onion, garlic, cabbage, tobacco, and alcohol), refrain from using deodorant, perfume or scented lotion, and had to bathe with a perfume-free soap that the experimenters provided. The morning before sweat collection, they had to rinse their armpits with clear water only.

Each participant watched two different videos on two different days (in a counterbalanced order), while wearing cotton pads under their armpits (7.5 × 7.5 cm, Euromedis, Neuilly-sous-Clermont, France) and a provided cotton t-shirt previously washed with clear water at 40°C. The video clip selected for the fear condition was a 20-minute excerpt from the movie *Sinister* [43]. The clips selected for the joy condition were the sketch *Les inventions* by Florence Foresti (a famous French humourist), *Singing in the rain*'s dance scene, *Have you ever seen a giraffe with a necklace* (Chez Wam production), *The Willy Waller 2006* (tetesaclaques.tv), *Birds* (Pixar), and *We go together* from *Grease*; for a total of 20 minutes.

After each session, participants placed the cotton pads in airtight sealed bags, which were stored in a freezer at −70 °C until presentation to horses. Participants indicated in a questionnaire whether they had thoroughly followed the dietary and hygienic instructions, and rated on 7-point Likert scales their extent of feeling angry, fearful, happy, sad, disgusted, neutral, surprised, calm, and amused. Samples from 3 participants who had not followed the dietary and hygienic instructions and one who did not take part in both sessions were excluded. Among the remaining samples we selected the ones from 14 participants (2 males and 12 females) to match the number of horses included in the study (see Animals below). They were the 14 participants who experienced the most fear in the fear condition and the most joy in the joy condition (participants were sorted in ascending order according to their rating of feeling fearful in the fear condition and to their rating of feeling joyful in the joy condition; and the 14 best-ranking participants were selected – see Supplementary Information, Table S1 in S1 File). Ratings for all evaluated emotions were combined and averaged to create indicators of high arousal (angry, fearful, happy, disgusted, surprised, amused), low arousal (neutral, sad, calm), positive affect (happy, calm, amused), and negative affect (angry, fearful, sad, disgusted – [14]). These scores revealed notable distinctions between the contexts for the selected participants, primarily in positive and negative affect scores, with some marginal variations in low arousal scores as well (Table 1). At the time of odour collection, unused cotton pads were placed in airtight sealed bags inside a freezer at −70 °C to be used as control odours.

## Animals

The study involved 43 Welsh mares (*Equus caballus*) aged 7.9 ± 2.2 years (mean ± s.d.) reared and living at the Animal Physiology Experimental Unit of the Orfrasière (UEPAO, 37380 Nouzilly, France, https://doi.org/10.15454/1.5573896321728955E12), INRAE. These mares lived in groups in outdoor paddocks. Fodder and water were available ad libitum. These horses are used only for research purposes and are handled daily by humans. Horses were randomly assigned to one of three experimental groups: Joy group (n = 14), to be exposed to human odours from the joy context; Control group (n = 15), to be exposed to control odours (unused cotton pads), or Fear group (n = 14), to be exposed to human odours from the fear context.

## Procedure

**Odour apparatus.** In the morning, the cotton pads that were to be used during the day were transferred from the −70 °C to a −20 °C freezer that was close to the experimental area. The pads were thawed at room temperature (≈15°C) in the airtight bags 10 minutes before being used. The apparatus consisted of a disposable lycra muzzle (designed for this experiment) in which two cotton pads were stapled so that they would be in front of the horse's nostrils once the apparatus was in place (Fig 1). Horses could breathe normally through this muzzle (they did not show any change in respiratory rate or breathing movements). The cotton pads presented to a horse by this apparatus were either two unused cotton pads

**Table 1. Experience of participants during the fear and joy contexts, transformed into dimensions of core affect (median[Q1,Q3]).**

|  | High arousal | Low arousal | Positive affect | Negative affect |
|---|---|---|---|---|
| Fear context | 2.83[2.50,3.25] | 2.33[2.00,2.67] | 2.00[1.42,2.25] | 2.50[2.25,2.75] |
| Joy context | 2.67[2.50,2.83] | 3.17[2.75,3.58] | 5.67[5.08,3.00] | 1.00[1.00,1.00] |

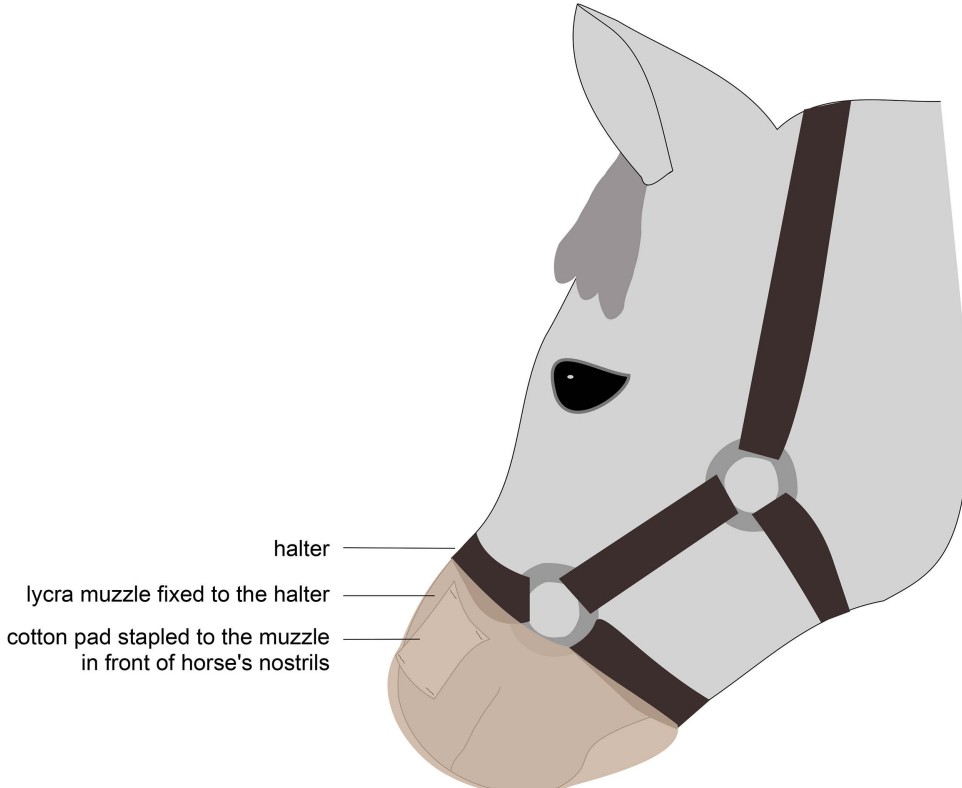

halter

lycra muzzle fixed to the halter

cotton pad stapled to the muzzle
in front of horse's nostrils

**Fig 1. Muzzle used to expose horses to the odours.** A lycra muzzle with cotton pads stapled in front of the horse's nostrils was fixed to the halter.

(Control group) or two cotton pads from a single participant in one of the emotional contexts (fear context for the Fear group, joy context for the Joy group). The muzzles (a new one for each horse) had been previously washed with clear water without detergent and were always manipulated wearing gloves.

**Behavioural tests.** The tests took place in a 2.7 m x 8.1 m box designed to carry out reactivity tests for horses [38]. It was divided into 12 zones of equal surface, marked by tape (Fig 2). An audience horse from the tested horse's herd was present at one end of the box, 1m from the fence, to avoid the tested horse to be stressed by a separation from conspecifics. She had been habituated to all the stimuli used in the tests and did not react to them. Four tests were conducted for each horse, two in the morning (morning series) and two in the afternoon (afternoon series), in a same order for all horses (from the less to the more frightening to the horse). For each test series, the cotton pads used were freshly thawed from the −20 °C freezer, with pads from the same donor used for a given horse in the morning and afternoon (each donor produced 4 pads, two of which were used for one horse in the morning and the other two for the same horse in the afternoon). On each testing day, the same number of horses were picked out randomly from each experimental group (Joy, Control or Fear) to be tested. Right before a series of tests began, the experimenter, who was familiar to the horses, led the horse from her paddock to the box, while two assistants prepared the odour apparatus (stapling the cottons pads to the muzzle) on a table a few meters from the test box. Only one of these assistants (hereby, the first assistant) was aware of which group the horse belonged to (Joy, Control or Fear), while the experimenter and the second assistant were both blind to the condition. Before entering the box, the muzzle with the odours was placed on the horse's mouth and nostrils (Fig 1). The horse was then set free in the box for 1 minute (habituation phase). Then, the series of two tests were conducted. The second assistant, who was unaware of the horse's group, recorded the

horse's behaviours live from outside the box, and the experiment was filmed by three cameras so that their notes could be completed afterwards if necessary. The morning series consisted of a grooming test and a free human approach test. The afternoon series consisted of a suddenness test and a novel object test.

**Grooming test:** The experimenter entered the box, tied a leading rope to the horse's halter, and scratched her on the wither and neck in a standardized manner (see [44]) for two minutes (Fig 2a). The number of times the horse touched the experimenter was recorded, as a measure of her willingness to further interact with the experimenter.

**Free human approach test:** The experimenter set the horse free, walked to a predefined standing spot (Fig 2b), and stood for three minutes looking at her feet. The number of times the horse voluntarily touched the experimenter was recorded, as a measure of her willingness to interact with the experimenter.

**Suddenness test:** A hole was cut at the front of the muzzle so that the horse could eat (the cotton pads remained in place). A bucket with food pellets was placed on the ground on a predefined spot (Fig 2c). The first assistant placed a folded umbrella in a hole through the wall over the bucket, and waited for the horse to eat the food. Three seconds after the horse started eating, or after two minutes if the horse was still not eating from the bucket, the second assistant triggered the opening of the umbrella. The intensity of the startle response was recorded during the test as follows: 0 = no reaction, 0.025 = head raise, 0.25 = head raise and startle, 0.5 = step back, 1 = quarter turn (10°-90°, front legs <20 cm above ground), 1.5 = violent quarter turn (10°-90°, front legs >20 cm above ground) or half turn (90°-180°, front legs <20

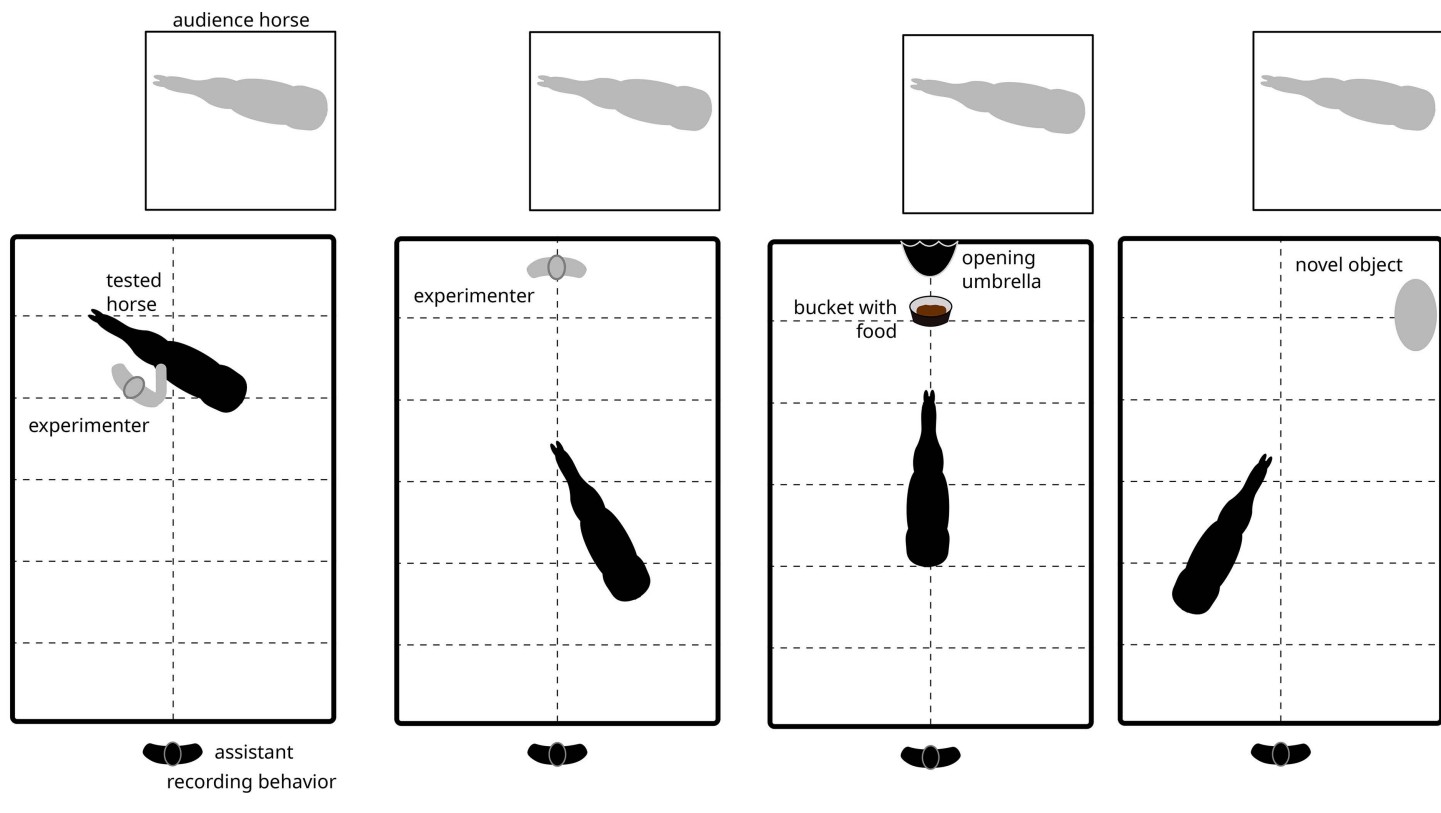

**Fig 2. Behavioural tests.** The experimental area was a 2.7 x 8.1 m box divided in 12 zones marked by tape. The audience horse was habituated to the stimuli, she could not touch or be touched by the tested horse. The tested horse was exposed to human odours from a joy or fear context or to control odours, according to her group.

cm above ground), 2=violent half turn (90°-180°, front legs >20 cm above ground), 2.5=very violent half turn (90°-180°, front legs >40 cm above ground, very rapid movement (<0.5s)), 3=fall [45].

**Novel object test:** A novel item, unfamiliar to the horse, was placed inside the box on a predefined spot (Fig 2d). The object was a composition crafted from linoleum pieces and variously coloured and shaped plastic pieces connected by strings, its overall dimensions were 1mx1m. The horse was set free in the box for three minutes. Two behaviours of the horse were recorded as measures of her fear toward the object [39,45]: the number of gazes at the object and the number of contacts the horse made with the object (touching the object with the nose or another body part)..

## Physiological measurements

Horses were habituated to the measurements from previous experiments.

**Salivary cortisol.** Saliva samples were collected right before and right after each series of test (15–20 minutes apart) with Salivette® Cortisol (Sarstedt, Nümbrecht, Germany). Cotton buds were centrifuged at 3000×g for 20 min at 4 °C, and the saliva was stored at −20 °C until analysis. The cortisol was measured in two 50 µL saliva aliquots per sample, using a competitive enzyme immunoassay in the INRAE Laboratory, Nouzilly, France. The assay sensitivity was 2ng/mL. For each sample the mean concentration of cortisol in the two aliquots was calculated. Then, for each horse the variation in salivary cortisol was calculated as the difference between the mean level before and after the test series.

**Heart rate.** Horses were equipped with a heart rate monitor (Polar Equine RS800CX Science, Polar Oy, Finland), recording RR data during the human approach, suddenness, and novel object tests (in the grooming test, grooming movements on the horse's wither prevented the monitor from functioning normally). RR data were then extracted from the recordings and converted to heart rate values. A visual inspection of this data was performed to detect artefactual beats (as recommended by [46]). Numerous artefacts were spotted, so data were cleared up based on the following algorithm (code used to apply these rules is available in the online repository, see Figure S1 in S1 File for more details): for each data point, if it had a difference of more than 35 bpm with the previous point, it was considered artefactual and removed from the dataset. If for a horse more than 30% of points concerning a test were artefactual, these data were considered unreliable and all data concerning this test for this horse were removed from the dataset. This was repeated three times. The number of horses whose heart rate data were included in the analysis for each test are reported in Supplementary Information – Table S2 in S1 File, representing 72% to 90% of the initial sample. Mean and maximum heart rate were then calculated for each test.

## Precautions to limit odour contaminations

The odour samples were always manipulated with disposable gloves and an FFP2 mask. In order to limit the spread of her own body odour, the experimenter took two showers a day (one before each testing half-day) using unscented soap, and wore a swimming cap during the tests. She and the two assistants did not wear any perfume or deodorant for the time of the experiment. She wore gloves, an FFP2 mask and clean overalls that were changed for each horse, and boots that were cleaned with clear water between each horse. The faeces was picked up and the box was thoroughly cleaned with a water jet after each horse finished the tests. The muzzle used for each horse was disposed of after each horse finished the tests.

## Statistical analysis

All statistical analysis were performed using R 4.3.1 [47], and figures were generated using the packages *ggplot2* [48] and *factoextra* [49]. The significance threshold was set at $\alpha \leq 0.05$. As this was a pioneering study exploring a poorly understood phenomenon, we aspired to detect any effect of human emotional odors on horses' behavior and physiology and also considered marginally significant results, with a criterion of p-values $\leq 0.1$. Tukey's method was employed to adjust p-values when performing multiple comparisons.

**Behavioural variables.** To test whether horses reacted differently to the tests according to the odour they smelled, we explored each recorded behaviour with generalized linear mixed models (GLMM) from the package *glmmTMB* [50], using linear and Poisson distributions as appropriate for the given variable (see Table S3 in S1 File). For each variable, a model was constructed to assess the effect of the group (Joy, Control or Fear) on the behaviour. Odour donor identity was added as a random factor to account for individual factors potentially influencing emotional odours (sex, age, genetics, etc). The models were compared to the null model via likelihood-ratio (chi-square) tests (results of these tests are given in Supplementary Information – Table S3 in S1 File). Distributions, within-group variance and homoscedasticity of the residuals were checked using the package *DHARMa* [51]. A chi-square test (Type II Wald) was then performed on each model to assess the influence of the group on the response variable using the function *Anova()* from the package *car* [52]. Post hoc tests based on Tukey's method were performed when necessary with the function *emmeans()* from the package *emmeans* [53].

To visualize overall differences between groups across behaviours, a principal component analysis (PCA) was performed using the function *prcomp()* from the *factoextra* package [49]. Correlations between all the variables were checked using the function *corrplot()* from the package *corrplot* [54] (Figure S2 in S1 File). To test whether overall differences in behaviour differed significantly between groups, we explored the first two components of the PCA with generalized linear mixed models (GLMM), following the same method as for individual variables (i.e., we tested whether the coordinates of individuals along these first two components differed according to the group).

**Physiological variables.** The evolution of cortisol level (before/after each test series), and the mean and maximum heart rate were analysed using linear mixed models (LMM) from the package *glmmTMB* [50]. For each variable, a model was built to assess the effect of the group on the physiological parameter. Odour donor identity was added as a random factor to account for individual factors potentially influencing emotional odours (sex, age, genetics, etc). For the evolution of cortisol level, the series (morning or afternoon) was also added as a random effect, to assess for potential variability introduced by cortisol circadian rythms [55]. The models were compared to the null model via likelihood ratio (chi-square) tests (results of these tests are given in Supplementary Information – Table S4 in S1 File). Distributions, within-group variance and homoscedasticity of the residuals were checked using the package *DHARMa* [51]. For models that were significantly better than the null model, a chi-square test (Type II Wald) was performed to assess the influence of the group on the response variable using the function *Anova()* from the package *car* [52]. Post hoc tests based on Tukey's method were performed when necessary with the function *emmeans()* from the package *emmeans* [53]. Moreover, a post-hoc power analysis was performed for the evolution of cortisol level with the function *pwr.t.test()* from the package *pwr*.

## Results

Descriptive statistics and effect sizes are given for all variables in Supplementary Information – Table S5 in S1 File.

### Behavioural variables

The number of touching the experimenter during the human approach test and across all tests, along with the startle intensity in the suddenness test and the number of gazes toward the novel object differed significantly according to the group (Table 2). Post-hoc tests revealed horses in the Fear group touched the experimenter less and showed higher fear responses compared to the Control and/or the Joy groups, according to the variable (Table 2 and Fig 3). The number of touching the experimenter during the grooming test and the number of contacts with the novel object differed marginally significantly between groups (Table 2). Post-hoc tests revealed that horses in the Fear group touched the experimenter marginally significantly less and made significantly less contacts with the novel object compared to the Joy group (Table 2 and Fig 3).

The two first factors of the PCA explained 62.4% of the total variability (Table 3 and Fig 4). Coordinates along the first factor (F1) differed significantly according to the group, post-hoc tests revealed horses in the Fear group had lower

**Table 2. Results of the tests performed for each behavioural variable.**

| Test | Variable | Overall group effect | | Post-hoc tests | | |
|------|----------|----------------------|------|----------------|---|---|
| | | Chi-square test | Df | | | |
| Grooming | Number of touching the experimenter | χ²=5.24  **p=0.072°** | 2 | Joy-Fear | Z=2.26 | **p=0.062°** |
| | | | | Joy-Control | Z=0.57 | p=0.84 |
| | | | | Control-Fear | Z=1.76 | p=0.18 |
| Human approach | Number of touching the experimenter | χ²=6.91  **p=0.032*** | 2 | Joy-Fear | Z=2.55 | **p=0.029*** |
| | | | | Joy-Control | Z=0.52 | p=0.86 |
| | | | | Control-Fear | Z=2.12 | **p=0.086°** |
| Suddenness | Intensity of startle | χ²=11.04  **p=0.004**** | 2 | Joy-Fear | t=−2.31 | **p=0.066°** |
| | | | | Joy-Control | t=0.88 | p=0.66 |
| | | | | Control-Fear | t=−3.23 | **p=0.0071**** |
| Novel object | Number of gazes | χ²=7.20  **p=0.027*** | 2 | Joy-Fear | Z=2.52 | **p=0.032*** |
| | | | | Joy-Control | Z=−0.67 | p=0.78 |
| | | | | Control-Fear | Z=−1.91 | p=0.14 |
| | Number of contacts | χ²=5.84  **p=0.054°** | 2 | Joy-Fear | Z=2.13 | **p=0.043*°** |
| | | | | Joy-Control | Z=0.26 | p=0.96 |
| | | | | Control-Fear | Z=1.98 | p=0.12 |
| PCA | F1 | χ²=23.4  **p<0.001**** | 2 | Joy-Fear | t=4.54 | **p=0.0002***** |
| | | | | Joy-Control | t=0.85 | p=0.67 |
| | | | | Control-Fear | t=3.76 | **p=0.0016**** |
| | F2 | χ²=0.45  p=0.80 | 2 | | | |

°p ≤ 0.1, *p ≤ 0.05, **p ≤ 0.01, ***p ≤ 0.001. Joy: horses exposed to cotton pads from humans watching a joyful video, Control: horses exposed to unused cotton pads, Fear: horses exposed to cotton pads from humans watching a horror video. Results of comparisons to the null model are available in Supplementary Information – Table S3 in S1 File.

coordinates along this factor (they were more on the left of the graph) compared to the Joy and Control groups (Table 2). This factor (F1) corresponded to horses' willingness to interact with the experimenter (positive values: willing to interact) and to the intensity of fear reactions (negative values: stronger fear reactions).

## Physiological variables

The maximum heart rate in the suddenness test differed significantly according to the group (Table 4). Post-hoc tests revealed that the maximum heart rate of horses in the Fear group was significantly higher than that of horses in the Joy group and in the Control group (Fig 3f, Joy-Fear: t=−2.66, p=0.034, Joy-Control: t=0.32, p=0.94, Fear-Control: t=−2.99, p=0.016). The other cardiac variables did not differ significantly according to the group (Table 4).

The variation in cortisol level showed marked variability in the sample (Mean±s.e., Joy: 0.86 ± 1.79, Neutral: 1.18 ± 1.76, Fear: 0.78 ± 1.73) with no significant effect of the group (Table 4). Post-hoc power analysis revealed a power of 0.15, indicating that variability in the sample was too high considering the sample size to detect a significant effect in this study.

## Discussion

In this study, we showed that horses' behaviour and physiology were influenced when exposed to odours emitted by humans when feeling fear or joy. When exposed to the odour from the fear context, horses touched the experimenter less, gazed more at the novel object, and were more startled in the suddenness test, compared to horses exposed to odours from the joy context or control odour. These behaviours indicate higher fear responses [39,40,45] and less willingness to interact with humans [41] when exposed to fear-related odours. Principal component analysis confirmed that horses were

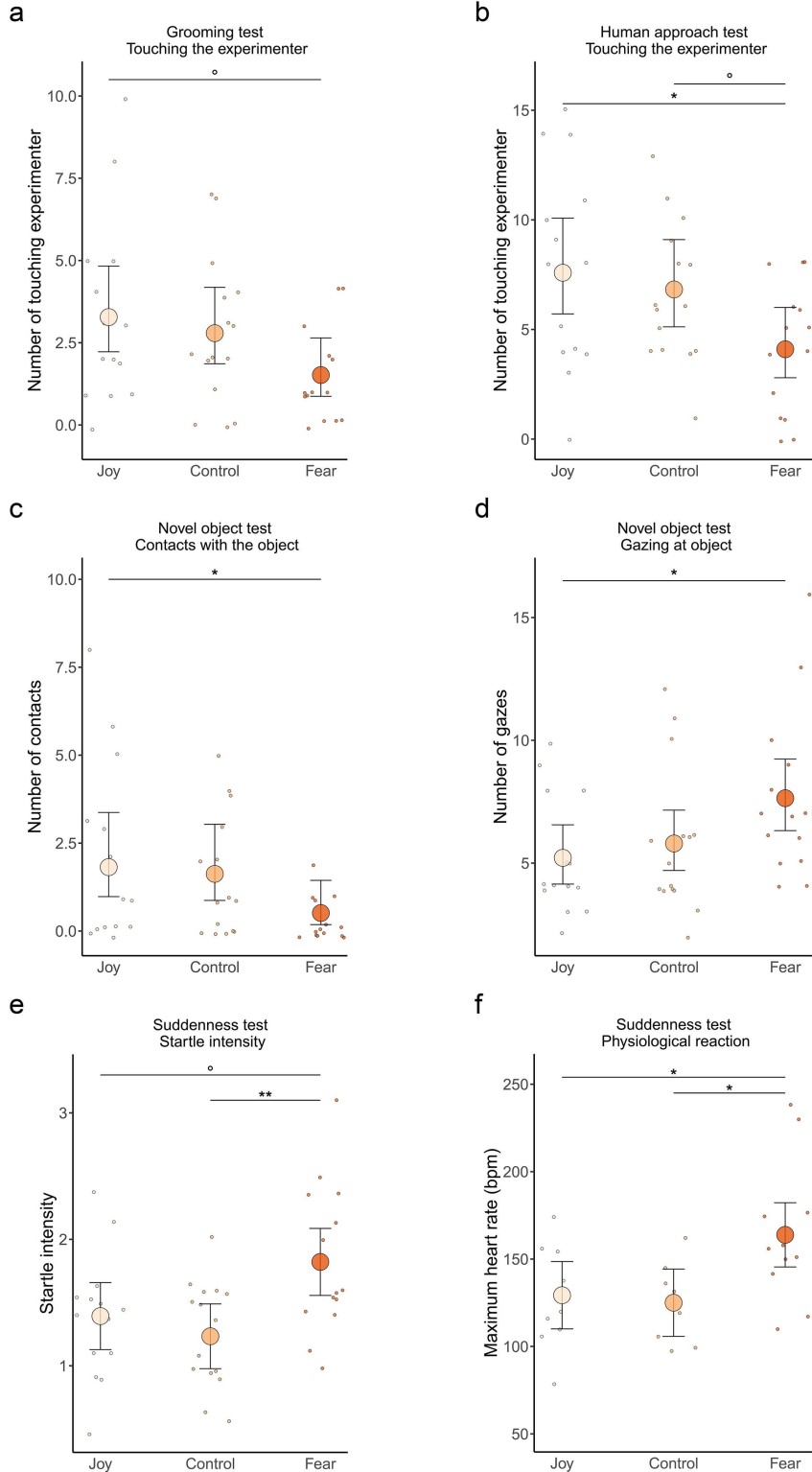

**Fig 3. Behaviour and physiology of horses during the tests according to the odour smelled. a.** Number of touching the experimenter in the Grooming test **b.** Number of touching the experimenter in the Free human approach test, **c.** Number of contacts with the objects in the Novel object test, **d.** Number of gazes at the object in the Novel object test, **e.** Startle intensity in the Suddenness test, **f.** Maximum heart rate in the Suddenness test. Joy:

horses exposed to cotton pads from humans watching a joyful video, Control: horses exposed to unused cotton pads, Fear: horses exposed to cotton pads from humans watching a horror video. Individual data points with mean and standard error from the models presented in Supplementary Information, Tables S3 and S4 in S1 File. °p ≤ 0.1, *p ≤ 0.05, **p ≤ 0.01.

**Table 3. Correlations between the variables included in the PCA and the first two factors.**

| Test | Variable | F1 (37.1%) | F2 (25.3%) |
|---|---|---|---|
| Grooming | Number of touching the experimenter | 0.48 | −0.54 |
| Human approach | Number of touching the experimenter | 0.59 | −0.13 |
| Suddenness | Intensity of startle | −0.41 | −0.50 |
| Novel object | Number of gazes | −0.28 | 0.38 |
| | Number of contacts | 0.42 | 0.54 |

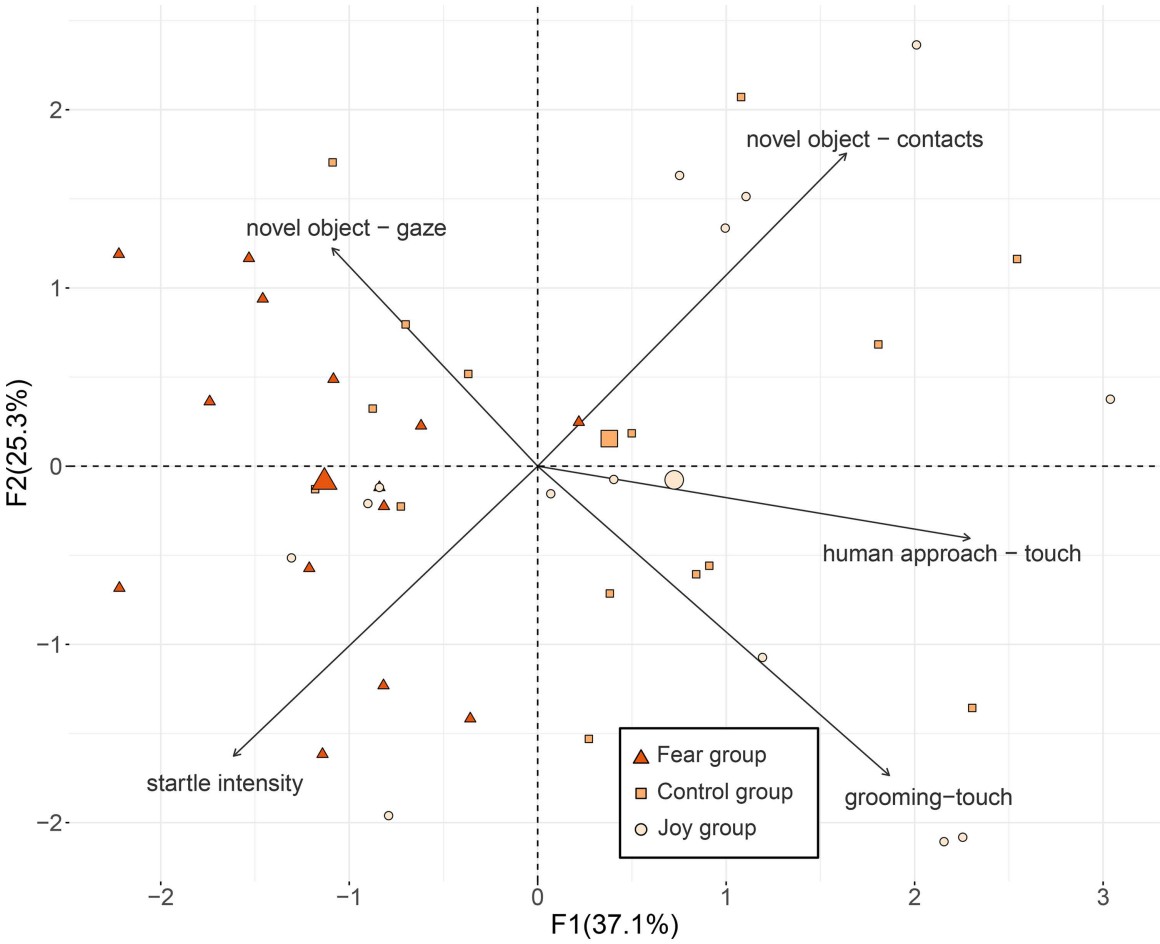

**Fig 4. Horses' behaviour during the tests as described by the PCA.** Fear group: horses exposed to cotton pads from humans watching a horror video, Control group: horses exposed to unused cotton pads, Joy group: horses exposed to cotton pads from humans watching a joyful video. The large points represent the barycentre for each group and the small points individual data points from the PCA.

Table 4. Results of the chi-square tests for each physiological variable (** p ≤ 0.01).

| Test | Variable | Overall group effect | | Df |
| --- | --- | --- | --- | --- |
| | | χ² | p value | |
| Human approach | Mean heart rate | 0.58 | 0.75 | 2 |
| | Maximum heart rate | 0.74 | 0.69 | 2 |
| Suddenness | Mean heart rate | 1.84 | 0.40 | 2 |
| | Maximum heart rate | 10.94 | 0.0042** | 2 |
| Novel object | Mean heart rate | 2.99 | 0.22 | 2 |
| | Maximum heart rate | 1.55 | 0.46 | 2 |
| All tests | Variation in cortisol level (before/after test series) | 0.99 | 0.61 | 2 |

**p ≤ 0.01. Results of comparisons to the null model are available in Supplementary Information – Table S4 in S1 File.

overall more afraid and less willing to approach humans when exposed to fear context odours, as their coordinates on the first axis were significantly lower in the Fear group compared to the Joy and Control groups.

More than just a reaction to odours, there appears to be an analogy between the emotional state of the donor (human) and that of the receiver (horse). Indeed, the observed responses indicate a shift towards a more negative and higher arousal emotional state in horses, consistent with the emotional valence and arousal of the human emitters. Correspondence in emotional valence and arousal between an emitter and receiver corresponds to criteria of emotional contagion defined as the transmission of an emotional state between two individuals [56,57]. Therefore, our observations suggest that emotional contagion of fear through odours, previously observed among humans [58–60], could also occur interspecifically, although these results still need validation from independent studies. The reactions observed in horses could be spontaneous, deriving from autonomic pathways connecting olfactory receptors to emotional reactions and behaviours; or they could result from associations formed by horses between some odours and the situations in which they are usually present (for example, horses could have associated odours emitted by fearful humans to stressful contexts). Indeed, horses are known to associate some characteristics of human stimuli (like the nature of a voice) to the valence of co-occurring interactions [61].

The fact that different species appear to respond to each other's emotional chemosignals is interesting, as emotions have previously been viewed as mainly internal states that serve to regulate individual behaviours [62,63] and produce signals that inform conspecifics of danger or resources [57,64]. The fact that these signals transcend species boundaries [18–21,36] suggests that they could also play a role in interspecific interactions, especially between humans and domestic mammals. Fear derives from the perception of danger [65], and is probably the most widely shared emotion in the animal kingdom. Its function is to trigger a fight-or-flight response [65], so emotional contagion of fear could be adaptive for animals to warn each other of danger, even when it involves different species. Olfactory signals provoking emotional contagion of fear appear particularly effective as odours remain effective in darkness, in the presence of obstacles, and persist even after the emitter has departed [1].

To date, responses to interspecific emotional signals have been observed mainly in domestic animals. For example, horses but also dogs, cats, cattle and goats can discriminate human emotional expressions from visual, auditory or olfactory signals [66]. Goats [67], horses [34] and cats [68] show preferences for positive emotions over negative ones, and dogs [69], horses [33,34], and cats [70] may even form mental representations of human emotions. This raises the question of whether these capacities emerged from domestication. Indeed, domestication has led to the "domestication syndrome," featuring traits like larger body size, increased reproduction, and reduced anti-predator behaviours [71,72], possibly including enhanced socio-cognitive capacities toward humans [73,74] and the ability to react to human emotional odours. However, domestic species do not necessarily surpass their wild counterparts in socio-cognitive abilities

[72], potentially due to the ancestral selection of wild species based on these traits. Specifically, regarding olfaction, responses to human emotional odours have been observed primarily in domesticated animals (horses [36,37] and dogs [19,20,75,76]), which could be explained by their ancestral species already having a particular aptitude for this. However, it is also possible that mammals possess receptors inherited from a common ancestor that allow detection of common molecules produced in sweat across species.

In this study, we did not find significant differences in horses' behaviour between odours from the joy context and unused cotton pads. Future studies employing habituation-discrimination tests similar to those conducted previously [37] could further elucidate horses' ability to differentiate between joy and control odours and explore their potential effects on horses' emotional states. In humans, odours from a joy context have been found to induce a facial expression and a way of looking at images indicative of happiness in the receivers [14], suggesting that these odours differ in nature to control odours and could also influence the behaviour of individuals from other species. Although it is less evident than for fear, reacting to interspecific signals of positive emotions could be adaptive. For species that have regular interspecific interactions (like domestic animals or species involved in mutualistic relationships) it could help to promote social bonds [57]. Moreover, happiness is part of the reward acquisition system, of which the role is to promote reward-seeking behaviours like feeding [62], and emotional contagion of happiness could encourage resource identification. However, positive chemosignals may be less salient than negative ones, or "joy" may have been too complex as an interspecific cue compared to "fear", to trigger observable differences in behaviour and physiology in the present study.

## Limitations and perspectives

In this study, no significant differences in cortisol level variations were observed, which limits the range of physiological consequences of human emotional odours on horses' physiology. These results could be due to circadian rhythms inducing marked variability [55], or to this parameter varying over longer periods than our tests allowed (less than an hour per horse). Therefore, it would be interesting to repeat this investigation over more extended periods [77,78]. On the other hand, increased maximum heart rate during the suddenness test suggests that physiological responses to interspecific fear-related odours do occur, which is in line with the function of emotions to recruit appropriate physiological resources in response to a stimulus, supporting survival [79]. However, the limited quality of heart rate data, and consequent data exclusion that was needed, encourage caution regarding these conclusions.

Another limitation is the animal sample, which included only females of Welsh breed. Other breeds and sexes (males, geldings) could react differently to human emotional odours, although these factors have not been identified as significant regarding horse-human interactions nor horse olfactory perception in previous studies (e.g., [25,29,30,33,35,36]). Moreover, the horses were all aged 5–12 years, and it would be interesting to investigate the influence of age on the observed phenomenon through a larger age sample, as this factor has been shown to influence horses' perception of human cues [80].

Regarding our experimental set-up, an audience horse was present for ethical reasons (preventing stress from social isolation), and in spite of all the precautions taken to limit her influence on the results (same audience horse across groups, habituated to the set-up and stimuli, her fence 1 m to the test box), consequences of her presence on the focal horse cannot be formally excluded. Further studies on horses' response to human emotional odours in different contexts will be valuable to confirm the observed phenomenon.

Last, recent studies have started to identify chemical families and compounds that might be responsible for the emotional content of human body odours [15,16]. Therefore, in addition to selecting samples from human donors who were in the intended emotional states, which could be a source of bias, it would be interesting to analyze the chemical content of odours presented to horses or other animals in behavioural studies like the present. Moreover, previous studies on human body odours revealed potential differences in nature and quantity of emotional body odours produced by men and women [81,82], and other parameters such as genetics, age or physical condition are also known to influence body

odours [83–85]. Although donor identity was included as a random factor in our study to account for such potential variations, it would be interesting to determine in further studies the influence of these parameters on emotional communication through odours in human-animal interactions.

Overall, the relatively innovative character of this study and field of investigation makes our inferences subject to potential adjustments and precision from further studies conducted in other contexts.

### Practical implications

Beyond their theoretical contribution to understanding interspecific emotional communication, these findings may have practical implications. In equine management and training contexts, human emotional states could directly influence horses' behavioural and physiological responses, particularly when handlers experience fear or stress. This suggests that caretakers' and riders' emotional regulation could be an important component of equine welfare, safety, and training effectiveness. For instance, in equestrian sports, therapeutic riding programs, or clinical handling, human fear-related odours might inadvertently increase horses' reactivity, potentially raising the risk of accidents or impairing human–horse interactions. Recognizing this pathway opens new avenues for designing training programs that integrate human emotional awareness, developing strategies to reduce stress transmission between humans and horses, and selecting or conditioning horses according to their sensitivity to human chemosignals.

### Conclusion

In this study, we showed that human emotional body odours can influence horses' behaviour and physiology, with human sweat samples from a fear context inducing higher fear responses in horses and lower interactions with humans. This highlights the significance of chemosignals in interspecific interactions and provides new insights into questions about the impact of domestication on emotional communication. Practical implications include acknowledging the importance of handlers' emotional state and its potential transmission through chemosignals during human-horse interactions.

### Supporting information

**S1 File. Supplementary Tables and Figures (Tables S1 to S5, Figures S1&S2).**
(PDF)

### Acknowledgments

We thank the staff from UEPAO for technical help, and all human participants who watched the joy and fear videos and donated their sweat for the experiment.

**Use of Artificial Intelligence (AI) and AI-assisted technologies** During the preparation of this work the authors used ChatGPT and Mistral in order to improve readability and language. After using this tool, the authors reviewed and edited the content as needed and take full responsibility for the content of the publication.

### Author contributions

**Conceptualization:** Alexandra Destrez, Fabrice Damon, Fabrice Reigner, Vitor H. B. Ferreira, Ludovic Calandreau, Léa Lansade.

**Data curation:** Noa Tanguy-Guillo, Céline Parias.

**Formal analysis:** Plotine Jardat, Noa Tanguy-Guillo.

**Funding acquisition:** Plotine Jardat.

**Investigation:** Plotine Jardat, Noa Tanguy-Guillo, Anne-Lyse Lainé, Céline Parias.

**Methodology:** Plotine Jardat, Alexandra Destrez, Fabrice Damon, Noa Tanguy-Guillo, Anne-Lyse Lainé, Céline Parias, Fabrice Reigner, Vitor H. B. Ferreira, Ludovic Calandreau, Léa Lansade.

**Project administration:** Plotine Jardat, Léa Lansade.

**Resources:** Fabrice Reigner, Léa Lansade.

**Software:** Plotine Jardat.

**Supervision:** Alexandra Destrez, Fabrice Damon, Vitor H. B. Ferreira, Ludovic Calandreau, Léa Lansade.

**Validation:** Anne-Lyse Lainé.

**Visualization:** Plotine Jardat.

**Writing – original draft:** Plotine Jardat.

**Writing – review & editing:** Alexandra Destrez, Fabrice Damon, Vitor H. B. Ferreira, Ludovic Calandreau, Léa Lansade.

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
