## [Decision Letter · Decision Letter 0]

15 Sep 2025

PONE-D-25-38894Human emotional odours influence horses’ behaviour and physiologyPLOS ONE

Dear Dr. Jardat,

I have consulted with two researchers whose expertise closely aligns with your manuscript topic. Both reviewers acknowledge that your study is innovative and has the potential to enhance the field of human-animal interaction research. However, Reviewer 2 raises important concerns about the transparency of your methods and limitations, as well as your interpretations of the results (that do not align with your actual findings). They also encourage you to consider the practical applications of your research in greater depth. Meanwhile, Reviewer 1 has provided some minor grammatical suggestions that can be addressed during your revision process to incorporate their feedback.

We look forward to receiving your revised manuscript.

Kind regards,

Brittany N. Florkiewicz, Ph.D.

Academic Editor

PLOS ONE

Journal Requirements:

2. Please expand the acronym “ANR and IFCE” (as indicated in your financial disclosure) so that it states the name of your funders in full.

Reviewers' comments:

Reviewer's Responses to Questions

**Comments to the Author**

1. Is the manuscript technically sound, and do the data support the conclusions?

Reviewer #1: Yes

Reviewer #2: Yes

2. Has the statistical analysis been performed appropriately and rigorously? 

Reviewer #1: Yes

Reviewer #2: Yes

3. Have the authors made all data underlying the findings in their manuscript fully available?

Reviewer #1: Yes

Reviewer #2: Yes

4. Is the manuscript presented in an intelligible fashion and written in standard English?

Reviewer #1: No

Reviewer #2: Yes

5. Review Comments to the Author

Reviewer #1: PLOS ONE Manuscript Number: PONE-D-25-38894

Human emotional odours influence horses’ behaviour and physiology

I recommend publication of the manuscript after minor revision

Comments

The text needs some improvement. Some of the mistakes are mentioned below.

Line 113 Human axillary sweat odour was collected from 30 adult participants. Were all participants women? If yes, mention it in the text. If no, mention how men and women were distributed. (I assume men and women smell different, especially in frightening situations.)

Line 186-187 Change ‘second assistant were blind to the condition’ to ‘second assistant was blind to the condition’

Line 193 Change ‘in’ to ‘of’

Line 221 Change ‘were’ to ‘was’

Line 252 Change ‘were’ to ‘was’

Line 285 Change ‘an chi-square test’ to ‘a chi-square test’

Line 357 Change ‘on’ to ‘of’

Line 395-389 The sentence is a bit long. Change to ‘However, results indicate a gradient in horses’ reactions across the Joy, Control, and Fear groups, as depicted in graphical representations (Fig3a-d&f); and the detection of significant differences between the Fear group and the Joy group but not between the Fear group and the Control group. This result suggests that horses’ reaction to the control odours was not equivalent to that to that of the odour from the joy context.’

Line 404 Change sentence ‘…suggests physiological responses to interspecific fear-related odours do occur,’ to ‘…suggests that physiological responses to interspecific fear-related odours do occur,’

Reviewer #2: Dear Authors,

This study addresses important and increasingly relevant topic in animal cognition and human–animal interaction, investigating the role of human emotional chemosignals in interspecific communication with horses.

Overall, the study is well-conceived and conducted, and most results support the authors’ hypotheses, providing novel perspectives on cross-species emotional transmission. However, several aspects require substantial clarification and revision before the manuscript can be considered for publication.

Major Revisions Required

1. Clarification on Horses’ Responses to Joy Odours

The discussion states that horses showed a "gradient" of responses to joy, control, and fear odours, suggesting that control was not equivalent to joy. However, post hoc tests for individual behaviours (Table 2) do not show significant differences between Joy and Control in most variables, nor between Joy and Fear in some cases (e.g., startle intensity, number of contacts with the novel object). Request: Please revise the language regarding Joy odours. While the absence of significant differences is a valid result and the recommendation of future habituation–discrimination studies is appropriate, the discussion should be more cautious to avoid implying differences where statistical support is lacking. Consider offering a deeper explanation for the lack of clear behavioural response to Joy odours, possibly discussing the lower salience of positive chemosignals or the complexity of “joy” as an interspecific cue compared to “fear”.

2. Cortisol Limitations

The lack of significant differences in cortisol levels is an important limitation. Request: While the discussion acknowledges possible reasons (circadian rhythms, short testing duration), this limitation should be more explicitly integrated—either in the Conclusion of the abstract or in a dedicated “Limitations” paragraph within the Discussion. This would help readers properly interpret the scope of the physiological responses. Increasing the sampling period, as suggested, is a valuable recommendation for future studies.

3. Figure and Table Presentation

Figure 3: The caption is clear and the inclusion of individual data points is helpful. However, for some subplots (e.g., 3a, 3c), overlapping points make individual data hard to distinguish. Consider jittering or transparency to improve readability.

Table 2: For post hoc tests, consider also providing the overall F or χ² statistic of the full GLMM before the pairwise comparisons, to give readers an overview of the group effect before detailing pairwise contrasts. Currently, only the χ² from the null vs. full model is shown.

4. Methodological and Interpretative Transparency

Sample of Human Donors: Clarify the criteria used to select 14 donors out of the original 30. Animal Sample: Only Welsh mares (n=43) were tested. Explicitly discuss how this restricts generalization to other sexes, breeds, and ages. Audience Horse: The possible influence of an “audience horse” on focal horses’ behaviour is not mentioned as a limitation and should be acknowledged. Heart Rate Data Quality: A large proportion of artefacts were removed, but the exact percentage of excluded data per condition is not reported. This threatens reliability and must be stated.

5. Statistical Approach

The practice of considering p ≤ 0.1 as a “trend” without corrections for multiple comparisons could inflate false positives. Request: Justify this choice and consider using appropriate multiple comparison corrections (e.g., FDR, Bonferroni).

6. Reporting of Effect Sizes and Variability

The abstract presents the results without effect sizes or statistical values, which reduces transparency. Request: Provide effect sizes for key behavioural variables and more detail on cortisol variability. A power analysis for the non-significant cortisol results would be valuable.

7. Interpretation and Discussion

Some conclusions (e.g., emotional contagion) are overstated given the absence of chemical analyses of the odours and the lack of evidence on specific volatile compounds. Please tone down speculative statements and emphasize that these results are preliminary and context-specific. Explicitly expand the discussion of limitations: donor selection, restricted animal sample, data exclusion, absence of chemical identification of compounds. Clearly state that results may not generalize across breeds, sexes, or ages.

8. Practical Implications

The manuscript would benefit from a paragraph on the potential applied implications of these findings. For example:

Beyond their theoretical contribution to understanding interspecific emotional communication, these findings may have practical implications. In equine management and training contexts, human emotional states could directly influence horses’ behavioural and physiological responses, particularly when handlers experience fear or stress. This suggests that caretakers’ and riders’ emotional regulation could be an important component of equine welfare, safety, and training effectiveness. For instance, in equestrian sports, therapeutic riding programs, or clinical handling, human fear-related odours might inadvertently increase horses’ reactivity, potentially raising the risk of accidents or impairing human–horse interactions. Recognizing this pathway opens new avenues for designing training programs that integrate human emotional awareness, developing strategies to reduce stress transmission between humans and horses, and selecting or conditioning horses according to their sensitivity to human chemosignals.

Minor Revisions and Editorial Suggestions

9. Abbreviations: Ensure that all abbreviations (PCA, GLMM, LMM, etc.) are defined at first use in the main text.

10. Citation Consistency: Check consistency between in-text citations and the reference list.

11. Language and Style: The manuscript is clearly written and professional. A final careful proofreading for minor typographical or grammatical errors is still recommended.

Recommendation

This manuscript addresses an interesting research question with a solid methodological basis and valuable results, especially concerning fear odors. It is a valuable addition to the field of interspecific communication. Considering the strengths and the areas needing improvement, I recommend Major Revisions.

Sincerely,

6. PLOS authors have the option to publish the peer review history of their article (what does this mean? ). If published, this will include your full peer review and any attached files.

**Do you want your identity to be public for this peer review?** For information about this choice, including consent withdrawal, please see our Privacy Policy .

Reviewer #1: No

Reviewer #2: **Yes: ** Viviane Maria Oliveira dos Santos

---

## [Author Response · Author response to Decision Letter 1]

30 Oct 2025

Response to reviewers

Editor comments

I have consulted with two researchers whose expertise closely aligns with your manuscript topic. Both reviewers acknowledge that your study is innovative and has the potential to enhance the field of human-animal interaction research. However, Reviewer 2 raises important concerns about the transparency of your methods and limitations, as well as your interpretations of the results (that do not align with your actual findings). They also encourage you to consider the practical applications of your research in greater depth. Meanwhile, Reviewer 1 has provided some minor grammatical suggestions that can be addressed during your revision process to incorporate their feedback.

Dear editor,

Thank you for your comments on our manuscript. We have now revised the article, answering all the comments made by the two reviewers. Precisions have been added in the methods section and a new section stating the limitations and perspectives of our study has been added in the discussion. Practical implications have also been added in accordance with the suggestions of reviewer 2. Moreover, we carefully revised the grammatical issues raised by reviewer 1 and the whole manuscript was screened for any additional language errors. Last, following the reviewers’ comments, the statistical analysis was adjusted, with minor impacts on the results and no impact on our conclusions (see the manuscript with track changes where the differences are highlighted).

We hope that these modifications will satisfy you and the two reviewers and that our manuscript is now suitable for publication in your esteemed journal.

Reviewer 1

I recommend publication of the manuscript after minor revision

Thank you for taking the time to revise our manuscript, for your kind comments and for this recommendation. We modified the article following your remarks.

Comments

The text needs some improvement. Some of the mistakes are mentioned below.

Thank you for your vigilance. We have carefully revised the mentioned sentences and screened the whole manuscript for any other language error.

Line 113 Human axillary sweat odour was collected from 30 adult participants. Were all participants women? If yes, mention it in the text. If no, mention how men and women were distributed. (I assume men and women smell different, especially in frightening situations.)

Thank you for this question. The number of men and women is now mentioned in the methods section.

Line 115: “Human axillary sweat odour was collected from 30 adult participants (8 males and 22 females) who volunteered to take part in the experiment.”

Moreover, we agree, that sex, as well as other individual characteristics such as age, genetics, etc, could influence the smells of odor samples. As we used samples from each donor in each condition (fear and joy), the impact on our results should be limited. However, we have now included the identity of the donor in the statistical analysis to take this into account (with minor impact on the results).

Line 276 and 294: “Odour donor identity was added as a random factor to account for individual factors potentially influencing emotional odours (sex, age, genetics, etc).”

We also added a discussion about this in the new Limitations and Persepctives paragraph.

Line 454: “Moreover, previous studies on human body odours revealed potential differences in nature and quantity of emotional body odours produced by men and women [81,82], and other parameters such as genetics, age or physical condition are also known to influence body odours [83–85]. Although donor identity was included as a random factor in our study to account for such potential variations, it would be interesting to determine in further studies the influence of these parameters on emotional communication through odours in human-animal interactions.”

Line 186-187 Change ‘second assistant were blind to the condition’ to ‘second assistant was blind to the condition’

We meant that two people were blind to the condition: the second assistant and the experimenter. We modified the sentence to make this clearer.

Line 192: “Only one of these assistants (hereby, the first assistant) was aware of which group the horse belonged to (Joy, Control or Fear), while the experimenter and the second assistant were both blind to the condition.”

Line 193 Change ‘in’ to ‘of’

Thank you, this was modified according to your suggestion.

Line 221 Change ‘were’ to ‘was’

We meant that two behaviors were recorded, we modified the sentence to make this clearer.

Line 227: “Two behaviours of the horse were recorded as measures of her fear toward the object [39,45]: the number of gazes at the object and the number of contacts the horse made with the object (touching the object with the nose or another body part).”

Line 252 Change ‘were’ to ‘was’

Line 285 Change ‘an chi-square test’ to ‘a chi-square test’

Line 357 Change ‘on’ to ‘of’

Thank you, these sentences were modified according to your suggestions.

Line 395-389 The sentence is a bit long. Change to ‘However, results indicate a gradient in horses’ reactions across the Joy, Control, and Fear groups, as depicted in graphical representations (Fig3a-d&f); and the detection of significant differences between the Fear group and the Joy group but not between the Fear group and the Control group. This result suggests that horses’ reaction to the control odours was not equivalent to that to that of the odour from the joy context.’

Thank you for this suggestion. This sentence has now been removed from the manuscript following the comments of reviewer 2.

Line 404 Change sentence ‘…suggests physiological responses to interspecific fear-related odours do occur,’ to ‘…suggests that physiological responses to interspecific fear-related odours do occur,’

Thank you for this suggestion, we modified the sentence accordingly (this sentence is now part of another section following the comments of reviewer 2).

Line 428: “On the other hand, increased maximum heart rate during the suddenness test suggests that physiological responses to interspecific fear-related odours do occur, which is in line with the function of emotions to recruit appropriate physiological resources in response to a stimulus, supporting survival [79].” 

Reviewer 2

Dear Authors,

This study addresses important and increasingly relevant topic in animal cognition and human–animal interaction, investigating the role of human emotional chemosignals in interspecific communication with horses.

Overall, the study is well-conceived and conducted, and most results support the authors’ hypotheses, providing novel perspectives on cross-species emotional transmission. However, several aspects require substantial clarification and revision before the manuscript can be considered for publication.

Thank you for taking the time to revise our manuscript and providing useful comments to improve it. Please find below our answers to all your remarks. We hope that the major modifications we implemented now make our manuscript suitable for publication.

Major Revisions Required

1. Clarification on Horses’ Responses to Joy Odours

The discussion states that horses showed a "gradient" of responses to joy, control, and fear odours, suggesting that control was not equivalent to joy. However, post hoc tests for individual behaviours (Table 2) do not show significant differences between Joy and Control in most variables, nor between Joy and Fear in some cases (e.g., startle intensity, number of contacts with the novel object). Request: Please revise the language regarding Joy odours. While the absence of significant differences is a valid result and the recommendation of future habituation–discrimination studies is appropriate, the discussion should be more cautious to avoid implying differences where statistical support is lacking. Consider offering a deeper explanation for the lack of clear behavioural response to Joy odours, possibly discussing the lower salience of positive chemosignals or the complexity of “joy” as an interspecific cue compared to “fear”.

Thank you for this comment. We agree that the lack of significant differences in post hoc tests between Joy and Control, nor between Joy and Fear un some cases, make our conclusions on the differences between these odors weak. We removed the sentence about graphical differences and discussed the possible lower salience or complexity of positive chemosignals.

Line 418: “However, positive chemosignals may be less salient than negative ones, or “joy” may have been too complex as an interspecific cue compared to “fear”, to trigger observable differences in behaviour and physiology in the present study.”

2. Cortisol Limitations

The lack of significant differences in cortisol levels is an important limitation. Request: While the discussion acknowledges possible reasons (circadian rhythms, short testing duration), this limitation should be more explicitly integrated—either in the Conclusion of the abstract or in a dedicated “Limitations” paragraph within the Discussion. This would help readers properly interpret the scope of the physiological responses. Increasing the sampling period, as suggested, is a valuable recommendation for future studies.

Thank you for this remark. We added a discussion about the lack of significant differences in the cortisol analysis in a new Limitations and Perspectives paragraph (other issues are also discussed in this paragraph).

Line 424: “In this study, no significant differences in cortisol level variations were observed, which limits the range of physiological consequences of human emotional odours on horses’ physiology. These results could be due to circadian rhythms inducing marked variability [55], or to this parameter varying over longer periods than our tests allowed (less than an hour per horse). Therefore, it would be interesting to repeat this investigation over more extended periods [77,78].”

3. Figure and Table Presentation

Figure 3: The caption is clear and the inclusion of individual data points is helpful. However, for some subplots (e.g., 3a, 3c), overlapping points make individual data hard to distinguish. Consider jittering or transparency to improve readability.

Thank you for this suggestion, we added jittering and transparency to improve readability of this graph, all individual points should now be visible.

Table 2: For post hoc tests, consider also providing the overall F or χ² statistic of the full GLMM before the pairwise comparisons, to give readers an overview of the group effect before detailing pairwise contrasts. Currently, only the χ² from the null vs. full model is shown.

We are sorry that the tables were not clear enough. The χ² numbers provided in the main manuscript are indeed those of the full GLMM before the pairwise comparisons, while the χ² numbers for the null vs. full models are given in Supplementary Information. We have now made it clearer in the text and tables.

E.g., line 298: “The models were compared to the null model via likelihood-ratio (chi-square) tests (results of these tests are given in Supplementary Information - Table S3).”

And line 319:

Table 2: Results of the tests performed for each behavioural variable

Test Variable Overall group effect Post-hoc tests

Chi-square test Df

Grooming Number of touching the experimenter χ²=5.24 p=0.072° 2 Joy-Fear Z=2.26 p=0.062°

Joy-Control Z=0.57 p=0.84

Control-Fear Z=1.76 p=0.18

Human approach Number of touching the experimenter χ²=6.91 p=0.032* 2 Joy-Fear Z=2.55 p=0.029*

Joy-Control Z=0.52 p=0.86

Control-Fear Z=2.12 p=0.086°

Suddenness Intensity of startle χ²=11.04 p=0.004** 2 Joy-Fear t=-2.31 p=0.066°

Joy-Control t=0.88 p=0.66

Control-Fear t=-3.23 p=0.0071**

Novel object Number of gazes χ²=7.20 p=0.027* 2 Joy-Fear Z=2.52 p=0.032*

Joy-Control Z=-0.67 p=0.78

Control-Fear Z=-1.91 p=0.14

Number of contacts χ²=5.84 p=0.054° 2 Joy-Fear Z=2.13 p=0.043*°

Joy-Control Z=0.26 p=0.96

Control-Fear Z=1.98 p=0.12

PCA F1 χ²=23.4 p<0.001*** 2 Joy-Fear t=4.54 p=0.0002***

Joy-Control t=0.85 p=0.67

Control-Fear t=3.76 p=0.0016**

F2 χ²=0.45 p=0.80 2

° p≤0.1, * p≤0.05, ** p≤0.01, *** p≤0.001. Joy: horses exposed to cotton pads from humans watching a joyful video, Control: horses exposed to unused cotton pads, Fear: horses exposed to cotton pads from humans watching a horror video. Results of comparisons to the null model are available in Supplementary Information - Table S3.

4. Methodological and Interpretative Transparency

Sample of Human Donors: Clarify the criteria used to select 14 donors out of the original 30.

Thank you for this suggestion. We modified the text to make this clearer.

Line 138: “Among the remaining samples we selected the ones from 14 participants (2 males and 12 females) to match the number of horses included in the study (see Animals below). They were the 14 participants who experienced the most fear in the fear condition and the most joy in the joy condition (participants were sorted in ascending order according to their rating of feeling fearful in the fear condition and to their rating of feeling joyful in the joy condition; and the 14 best-ranking participants were selected – see Supplementary Information, Table S1).”

Animal Sample: Only Welsh mares (n=43) were tested. Explicitly discuss how this restricts generalization to other sexes, breeds, and ages. Audience Horse: The possible influence of an “audience horse” on focal horses’ behaviour is not mentioned as a limitation and should be acknowledged.

Thank you for these suggestions. A discussion about the sample and one about the audience horse were added in the new Limitations and Perspectives paragraph.

Sample, line 434: “Another limitation is the animal sample, which included only females of Welsh breed. Other breeds and sexes (males, geldings) could react differently to human emotional odours, although these factors have not been identified as significant regarding horse-human interactions nor horse olfactory perception in previous studies (e.g., [25,29,30,33,35,36]). Moreover, the horses were all aged 5 to 12 years, and it would be interesting to investigate the influence of age on the observed phenomenon through a larger age sample, as this factor has been shown to influence horses’ perception of human cues [80].”

Audience horse, line 441: “Regarding our experimental set-up, an audience horse was present for ethical reasons (preventing stress from social isolation), and in spite of all the precautions taken to limit her influence on the results (same audience horse across groups, habituated to the set-up and stimuli, her fence 1 m to the test box), consequences of her presence on the focal horse cannot be formally excluded. Further studies on horses’ response to human emotional odours in different contexts will be valuable to confirm the observed phenomenon.”

Heart Rate Data Quality: A large proportion of artefacts were removed, but the exact percentage of excluded data per condition is not reported. This threatens reliability and must be stated.

Thank you for this comment. This is now discussed in the Limitations and Perspectives paragraph.

Line 428: “On the other hand, increased maximum heart rate during the suddenness test suggests that physiological responses to interspecific fear-related odours do occur, which is in line with the function of emotions to recruit appropriate physiological resources in response to a stimulus, supporting survival [79]. However, the limited quality of heart rate data, and consequent data exclusion that was needed, encourage caution regarding these conclusions.”

5. Statistical Approach

The practice of considering p ≤ 0.1 as a “trend” without corrections for multiple comparisons could inflate false positives. Request: Justify this choice and consider using appropriate multiple comparison corrections (e.g., FDR, Bonferroni).

Thank you for raising this issue. We agree that considering results with 0.05 ≤ p ≤ 0.1 and testing multiple variables could inflate false positives. However, corrections such as Bonferroni are debatable as they induce an inflation of false negatives and could lead to overlook interesting results (see Streiner, 2015. Be

---

## [Decision Letter · Decision Letter 1]

16 Nov 2025

Human emotional odours influence horses’ behaviour and physiology

PONE-D-25-38894R1

Dear Dr. Jardat,

We’re pleased to inform you that your manuscript has been judged scientifically suitable for publication and will be formally accepted for publication once it meets all outstanding technical requirements.

Kind regards,

Brittany N. Florkiewicz, Ph.D.

Academic Editor

PLOS ONE

Additional Editor Comments (optional):

Reviewers' comments:

Reviewer's Responses to Questions

**Comments to the Author**

1. If the authors have adequately addressed your comments raised in a previous round of review and you feel that this manuscript is now acceptable for publication, you may indicate that here to bypass the “Comments to the Author” section, enter your conflict of interest statement in the “Confidential to Editor” section, and submit your "Accept" recommendation.

Reviewer #1: (No Response)

Reviewer #2: All comments have been addressed

2. Is the manuscript technically sound, and do the data support the conclusions?

Reviewer #1: Yes

Reviewer #2: Yes

3. Has the statistical analysis been performed appropriately and rigorously? 

Reviewer #1: I Don't Know

Reviewer #2: Yes

4. Have the authors made all data underlying the findings in their manuscript fully available?

Reviewer #1: Yes

Reviewer #2: Yes

5. Is the manuscript presented in an intelligible fashion and written in standard English?

Reviewer #1: Yes

Reviewer #2: Yes

6. Review Comments to the Author

Reviewer #1: Olfaction is "the oldest one of our senses" but little understood by us "micreomatic creatures". It is important both for safety reasons and for animal welfare reasons. For these reasons the study is of interest for both scientists as well as for riders and horse caretakers.

Reviewer #2: Dear Authors,

I have carefully examined the revised version of the manuscript entitled “Human emotional odours influence horses’ behaviour and physiology”. I appreciate the effort made to address all the comments and methodological clarifications requested in the first review.

The authors have provided detailed and satisfactory responses to each of the previous concerns. The new version includes clear justification for the selection of human donors, expanded discussion of the cortisol limitations, and improved transparency regarding the heart rate data quality. The statistical reporting has been refined with the inclusion of effect sizes and appropriate terminology regarding non-significant trends. Furthermore, the discussion now adopts a more cautious tone when interpreting emotional contagion and highlights the study’s limitations more explicitly.

The addition of the paragraph on practical implications greatly enhances the translational relevance of the findings, and the revised figures and tables are now clearer and easier to interpret. Language, formatting, and citation consistency have also improved substantially.

Overall, the revised manuscript is well written, methodologically sound, and provides a novel contribution to our understanding of interspecific emotional communication. I believe it now meets the publication standards of PLOS ONE.

Recommendation: Accept

7. PLOS authors have the option to publish the peer review history of their article (what does this mean? ). If published, this will include your full peer review and any attached files.

**Do you want your identity to be public for this peer review?** For information about this choice, including consent withdrawal, please see our Privacy Policy .

Reviewer #1: No

Reviewer #2: **Yes: ** Viviane Maria Oliveira dos Santos

---

## [Editor Report · Acceptance letter]

PONE-D-25-38894R1

PLOS ONE

Dear Dr. Jardat,

I'm pleased to inform you that your manuscript has been deemed suitable for publication in PLOS ONE. Congratulations! Your manuscript is now being handed over to our production team.

Kind regards,

on behalf of

Dr. Brittany N. Florkiewicz

Academic Editor

PLOS ONE